# Function Basis Encoding of Numerical Features in Factorization Machines

**Alex Shtoff**                                                    *alex.shtf@gmail.com*
*Yahoo*

**Elie Abboud**                                                    *eliabboud1000@gmail.com*
*Department of Computer Science*
*University of Haifa*

**Rotem Stram**                                                    *rotemstram@gmail.com*
*Yahoo*

**Oren Somekh**                                                    *orensomekh@yahoo.com*
*Yahoo*

**Reviewed on OpenReview:** *https://openreview.net/forum?id=M4222IBHsh*

## Abstract

Factorization machine (FM) variants are widely used for large scale real-time content recommendation systems, since they offer an excellent balance between model accuracy and low computational costs for training and inference. These systems are trained on tabular data with both numerical and categorical columns. Incorporating numerical columns poses a challenge, and they are typically incorporated using a scalar transformation or binning, which can be either learned or arbitrarily chosen. In this work, we provide a systematic and theoretically-justified way to incorporate numerical features into FM variants by encoding them into a vector of function values for a set of functions of one's choice.

We view FMs as approximators of *segmentized* functions, namely, functions from a field's value to the real numbers, assuming the remaining fields are assigned some given constants, which we refer to as the segment. From this perspective, we show that our technique yields a model that learns segmentized functions of the numerical feature spanned by the set of functions of one's choice, namely, the spanning coefficients vary between segments. Hence, to improve model accuracy we advocate the use of functions known to have  powerful approximation capabilities, and offer the B-Spline basis due to its well-known approximation power, widespread availability in software libraries and its efficiency in terms of computational resources and memory usage. Our technique preserves fast training and inference, and requires only a small modification of the computational graph of an FM model. Therefore, incorporating it into an existing system to improve its performance is easy. Finally, we back our claims with a set of experiments that include a synthetic experiment, performance evaluation on several data-sets, and an A/B test on a real online advertising system which shows improved performance. We have made the code to reproduce the experiments available at `https://github.com/alexshtf/cont_features_paper`.

## 1 Introduction

Traditionally, online content recommendation systems rely on predictive models to choose a set of items to display by predicting the affinity of the user towards a set of candidate items. These models are usually trained on feedback gathered from a log of interactions between users and items from the recent past. For systems such as online ad recommenders with billions of daily interactions, speed is crucial. The training

process must be fast to keep up with changing user preferences and quickly deploy a fresh model. Similarly, model inference, which amounts to computing a score for each item, must be rapid to select a few items to display out of a vast pool of candidate items, all within a few milliseconds. Factorization machine (FM) variants, such as Rendle (2010); Juan et al. (2016); Pan et al. (2018); Sun et al. (2021), are widely used in these systems due to their ability to train incrementally, and strike a good balance between being able to produce accurate predictions, while facilitating fast training and inference.

The training data consists of past interactions between users and items, and is typically given in tabular form, where the table's columns, or *fields*, have either categorical or numerical features. For example, "gender" or "time since last visit" are fields, whereas "Male" and "10 hours" are corresponding features. In recommendation systems that rely on FM variants, each row in the table is typically encoded as a concatenation of field encoding vectors. Categorical fields are usually one-hot encoded, whereas numerical fields are conventionally binned to a finite set of intervals to form a categorical field, and one-hot encoding is subsequently applied. A large number of works are devoted to the choice of intervals, e.g. Dougherty et al. (1995); Peng et al. (2009); Liu et al. (2002); Gama & Pinto (2006). Regardless of the choice, the model's output is a *step function* of the value of a given numerical field, assuming the remaining fields are kept constant, since the same interval is chosen independently of where the value falls in a given interval. For example, consider a model training on a data-set with "age", "device type", and "time the user spent on our site". For the segment of 25-years old users using an iPhone the model will learn some step function for different spending time values, whereas for the segment of 37-years old users using a laptop the model may learn a (possibly) different step function.

However, the optimal segmentized functions the model aims to learn, which describe the user behavior, aren't necessarily step functions. Typically, such functions are continuous or even smooth, and there is a gap between the approximating step functions the model learns, and the optimal ones. In theory, a potential solution is simply to increase the number of bins. This increases the approximation power of step functions, and given an infinite amount of data, would indeed help. However, the data is finite, and this can lead to a *sparsity* issue - as the number of learning samples assigned to each bin diminishes, it becomes increasingly challenging to learn a good representation of each bin, even with large data-sets, especially because we need to represent all segments simultaneously. This situation can lead to a degradation in the model's performance despite having increased the theoretical approximation power of the model. Therefore, there is a limit to the accuracy we can achieve with binning on a given data-set as demonstrated in Figure 1.

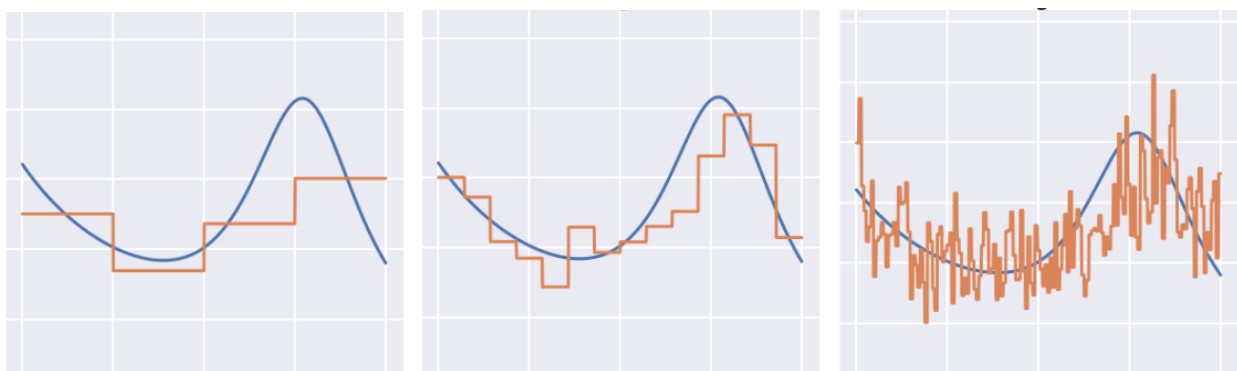

Figure 1: For a given segment, learned segmentized step function approximations (orange) of a true function (blue) which was used to generate a synthetic data-set. On the left - too few bins, "bad" approximation. In the middle - a balanced number of bins, "moderate" approximation. On the right - many bins, but approximation gets even "worse" due to a sparsity issue. Refer to Figure 4 for additional synthetic evaluation results with loss measurements.

In this work, we propose a technique to improve the accuracy of FM variants by reducing the approximation gap *without* sparsity issues, while preserving training and inference speeds. In other words, on the approximation-estimation balance, we aim to reduce both approximation and estimation error. Our technique is composed of encoding a numerical field using a vector of *basis functions*, and a minor modification to the computational graph of an FM variant. The idea of using nonlinear basis functions to model nonlinear functions is, of

course a standard practice with linear models. This work is about analyzing the *interplay* between nonlinear basis functions and the FM family, to yield a surprisingly powerful technique for recommender systems, and tabular datasets in general.

Indeed, we present an elementary Lemma showing that the resulting model learns a *segmentized* output function spanned by the chosen basis, meaning that spanning coefficients depend on the values of the remaining fields. This is, of course, an essential property for recommendation systems, since indeed users with different demographic or contextual properties may behave differently. For a pair of fields, the segmentized functions are spanned by a tensor product of the chosen pair of bases, where the coefficients are learned in *factorized* form by the model.

Based on the generic observation above, we offer the B-Spline basis (de Boor, 2001, pp 87) defined on the unit interval on uniformly spaced break-points, composed onto a transformation that maps the feature to the unit interval. The number of break-points (a.k.a knots) is a system hyper-parameter which can be further optimized. The strong approximation power of splines (de Boor, 2001, pp 149) ensures that we do not need a large number of break-points. Hence, we can closely approximate the optimal segmentized functions, without introducing sparsity issues. Moreover, to make integration of our idea easier in a practical production-grade recommendation system, we present a technique to seamlessly integrate a model trained using our proposed scheme into an existing recommendation system that employs binning, albeit with a controllable reduction in accuracy. Although a significant part of this work considers the B-Spline basis, the techniques we present can be used with an arbitrary basis.

Our work builds on the recent work of Rügamer (2022), that uses a similar technique in conjunction with FMs. Our work extends these results to a wider family of FM variants, and utilizes it from the perspective of recommender systems and a broaded practical scope: application to unbounded domains and interaction with categorical features, deployment in an existing recommender system, and A/B test results on a real-world online advertising product.

To summarize, the main contributions of this work are: (a) *Basis encoding* We propose introducing a slight modification to the computational graph of FM variants that facilitates encoding of numerical features as vector of basis functions; (b) *Spanning properties* We show that our modification makes any model from a family of FM variants learn *segmentized* functions spanned by pairwise tensor products of the basis of our choice, and inherits the approximation power of that basis; (c) *B-Spline basis* We justify the use of the B-Spline basis from both theoretical and practical aspects, and demonstrate their benefits via numerical evaluation; (d) *Ease of integration into an existing system* We show how to integrate a model trained according to our method into an existing recommender system which currently employs numerical feature binning, to significantly reduce integration costs; (e) *Simplified numerical feature engineering* the strong approximation power of the cubic B-Spline basis allows building accurate models without investing time and effort to manually tune bin boundaries, and use simple uniform break-points instead.

## 1.1 Related work

Putting aside the FM variants, there is a large body of work dealing with neural networks training over tabular data (Arik & Pfister, 2021; Badirli et al., 2020; Gorishniy et al., 2021; Huang et al., 2020; Popov et al., 2020; Somepalli et al., 2022; Song et al., 2019; Hollmann et al., 2022). Neural networks have the potential to achieve high accuracy and can be incrementally trained on newly arriving data using transfer learning techniques. Additionally, due to the *universal approximation* theorem (Hornik et al., 1989), neural networks are capable of representing a segmentized function from any numerical feature to a real number. However, the time required to train and make inferences using neural networks is significantly greater than that required for factorization machines. Even though some work has been done to alleviate this gap for neural networks by using various embedding techniques (Gorishniy et al., 2022), they have not been able to outperform other model types. As a result, in various practical applications, FMs are preferred over NNs. For this reason, in this work we focus on FMs, and specifically on decreasing the gap between the representation power of FMs and NNs without introducing a significant computational and conceptual complexity.

A very simple but related approach to ours was presented in Covington et al. (2016). The work uses neural networks and represents a numerical value $z$ as the triplet $(z, z^2, \sqrt{z})$ in the input layer, which can be seen

as a variant of our approach using a basis of three functions. Another approach which bears similarity to ours also comes from works which use deep learning, such as Cheng (2022); Gorishniy et al. (2021); Song et al. (2019); Guo et al. (2021). In these works, first-order splines are used in the input layer to represent continuity, and the representation power of a neural network compensates for the weak approximation power of first-order splines. Here we do the opposite - we use the stronger approximation power of cubic splines to compensate for the weaker representation power of FMs.

Finally, any comprehensive discussion on tabular data would be incomplete without mentioning *gradient boosted decision trees* (GBDT) (Chen & Guestrin, 2016; Ke et al., 2017; Prokhorenkova et al., 2018), which are known to achieve state-of-the-art results (Gorishniy et al., 2021; Shwartz-Ziv & Armon, 2022). However, GBDT models aren't useful in a variety of practical applications, primarily due to significantly slower inference speeds,namely, it is challenging and costly to use GBDT models to rank hundreds of thousands of items in a matter of milliseconds.

## 2 Background and problem statement

Factorization machines are formally described as models whose input is a feature vector $\boldsymbol{x}$ representing rows in tabular data-sets. In the context of recommendation systems, the data-set contains past interactions between users and items, whose columns, often named *fields*, and whose values are *features*. The columns describe the context, such as the user's gender and age, the time of visit, or the article the user is currently reading, whereas others describe the item, such as product category, or item popularity.

In this section we formally describe how we assume that the data is modeled as the feature vector $\boldsymbol{x}$, describe the FM family we shall focus on in this paper, and formally state the main problem we aim to solve.

### 2.1 Feature vector structure

Consider a tabular data-sets with $m$ fields $\{1, \ldots, m\}$, each can be either numerical or categorical field. We denote a row in the dataset with $(z_1, \ldots, z_m)$, each $z_i$ is a feature (value) associated with its corresponding field $f(i)$. Each feature $z_i$ is mapped to a vector of values by applying its corresponding encoding function $\mathbf{enc}_i(\cdot)$:

$$\boldsymbol{x} = \mathbf{enc}(z_1, \ldots z_m) \equiv \begin{bmatrix} \mathbf{enc}_1(z_1) \\ \vdots \\ \mathbf{enc}_m(z_m) \end{bmatrix},$$

For example, consider three fields: field 1 is the user's country, field 2 is the user's device type, and field 3 is the product category. All three are categorical. Then $\mathbf{enc}_1(\cdot)$ one-hot encodes the country, $\mathbf{enc}_2(\cdot)$ one-hot encodes the device, and $\mathbf{enc}_3(\cdot)$ one-hot encodes the product category. As a result, in this concrete example, $\boldsymbol{x}$ is the concatenation of three one-hot encodings.

### 2.2 Segmentized view of models

In this work we aim to study models learned on tabular data as a function of one or two tabular columns, while the remaining are kept constant. Therefore, we need some compact notation for the concept of viewing a multivariate function as a function of one or two variables, while keeping the rest constant.

For a given vector $\boldsymbol{y}$ we shall denote by $\boldsymbol{y}_{-k}$ the vector obtained from all its components except for $y_k$. For a function $\Phi(\boldsymbol{y})$, we shall denote by $\mathrm{seg}[\Phi; k](\boldsymbol{y}_{-k})$ the function obtained by assigning the values $\boldsymbol{y}_{-k}$ to all its arguments, except for $y_k$, producing a function of $y_k$. Formally, we have

$$\mathrm{seg}[\Phi; k](\boldsymbol{y}_{-k})(y_k) = \Phi(\boldsymbol{y}).$$

This "partial application" concept is not new. The mathematical term is *currying*, coined by Frege (1893). However, we shall use the term *segmenized view* of $\Phi$ for a reason that will become apparent immediately. Similarly to the exclusion of a single coordinate $k$, for a pair of coordinates $k, j$ we define $\boldsymbol{y}_{-(k,j)}$ and

$$\mathrm{seg}[\Phi; k, j](\boldsymbol{y}_{-(k,j)})(y_k, y_j) = \Phi(y).$$

Namely, $\text{seg}[\Phi; k, j](\boldsymbol{y}_{-(k,j)})$ assigns values to all inputs of $\Phi$, except for $y_k$ and $y_j$.

A machine learned model $\Phi(\boldsymbol{x})$ is a function of the *encoded* feature vector $\boldsymbol{x}$, but we are interested in studying it as a function of the original tabular columns. Thus, our aim is studying the composition

$$\Phi^{\text{enc}} = \Phi \circ \mathbf{enc}.$$

Given a field $f$, the function $\text{seg}[\Phi^{\text{enc}}, f](\boldsymbol{z}_{-f})$ is a function of $z_f$, given a concrete assignment $\boldsymbol{z}_{-f}$ to the remaining fields. Recalling our three-field example of country, device type, and product category, the function $\text{seg}[\Phi^{\text{enc}}, 2]("US", "Furniture")$ characterizes the model as a function of the device type for users from the US that interact with furniture products. From the recommender systems perspective, we are characterizing the model for a given *segment* of users and items, hence the name *segmentized* view, and the notation seg.

### 2.3 The factorization machine family

In this work we consider several model variants, which we refer to as the factorization machine family. The family includes the celebrated *factorization machine* (FM) (Rendle, 2010), the *field-aware factorization machine* (FFM) (Juan et al., 2016), the *field-weighted factorization machine* (FwFM) (Pan et al., 2018), and the *field-matrixed factorization machine* (FmFM) (Sun et al., 2021; Pande, 2021), that generalized the former variants. FMs are typically employed for supervised learning tasks on a dataset $\{(\boldsymbol{x}_t, y_t)\}_{t=1}^N$ with either binary or real-valued labels $y_t$.

The classical Factorization Machines (FMs), as proposed by Rendle (2010), compute

$$\Phi_{\text{FM}}(\boldsymbol{x}) = w_0 + \langle \boldsymbol{x}, \boldsymbol{w} \rangle + \sum_{i=1}^n \sum_{j=i+1}^n \langle x_i \boldsymbol{v}_i, x_j \boldsymbol{v}_j \rangle.$$

The learned parameters are $w_0 \in \mathbb{R}$, $\boldsymbol{w} \in \mathbb{R}^n$, and $\boldsymbol{v}_1, \ldots, \boldsymbol{v}_n \in \mathbb{R}^k$, where $k$ is a hyper-parameter. The model can be thought of as a way to represent the quadratic interaction model

$$\Phi_{\text{quad}}(\boldsymbol{x}) = w_0 + \langle \boldsymbol{x}, \boldsymbol{w} \rangle + \sum_{i=1}^n \sum_{j=i+1}^n A_{i,j} x_i x_j,$$

where the coefficient matrix $\boldsymbol{A}$ is represented in factorized form. The vectors $\boldsymbol{v}_1, \ldots, \boldsymbol{v}_n$ are the feature embedding vectors. Classical matrix factorization is recovered when we have only user id and item id fields, whose values are one-hot encoded.

The model $\Phi_{\text{FM}}$ does not represent the varying behavior of a feature belonging to some field when interacting with features from different fields. For example, genders may interact with ages differently than they interact with product categories. Initially, to explicitly encode this information into a model, the Field-Aware Factorization Machine (FFM) was proposed by Juan et al. (2016). Each embedding vector $\boldsymbol{v}_i$ is modeled as a concatenation of field-specific embedding vectors for each of the $m$ fields:

$$\boldsymbol{v}_i = \begin{bmatrix} \boldsymbol{v}_{i,1} \\ \vdots \\ \boldsymbol{v}_{i,m} \end{bmatrix},$$

where $\boldsymbol{v}_{i,f}$ is the embedding vector of feature $i$ when interacting with another feature from a field $f$. As a convention, we denote by $f(i)$ the field whose value is used to encode the $i^{\text{th}}$ component of the feature vector $\boldsymbol{x}$. The model computes

$$\Phi_{\text{FFM}}(\boldsymbol{x}) = w_0 + \langle \boldsymbol{x}, \boldsymbol{w} \rangle + \sum_{i=1}^n \sum_{j=i+1}^n \langle x_i \boldsymbol{v}_{i,f(j)}, x_j \boldsymbol{v}_{j,f(i)} \rangle$$

As pointed out by Pan et al. (2018); Juan et al. (2016), the FFM models are prone to over-fitting, since it learns a feature embedding vector for each feature $\times$ field pair. As a remedy, Pan et al. (2018) proposed the

Field-Weighted Factorization Machine (FwFM) that models the varying behavior of field interaction using learned scalar field interaction intensities $r_{e,f}$ for each pair of fields $e, f$. The FwFM computes

$$\Phi_{\text{FwFM}}(\boldsymbol{x}) = w_0 + \langle \boldsymbol{x}, \boldsymbol{w} \rangle + \sum_{i=1}^{n} \sum_{j=i+1}^{n} r_{f(i),f(j)} \langle x_i \boldsymbol{v}_i, x_j \boldsymbol{v}_j \rangle.$$

A generalization of all the models above has been independently proposed in Sun et al. (2021); Pande (2021). To describe it, let $\langle \boldsymbol{x}, \boldsymbol{y} \rangle_{\boldsymbol{P}} = \boldsymbol{x}^T \boldsymbol{P} \boldsymbol{y}$ denote the "inner product"[1] associated with some matrix $\boldsymbol{P}$. The FmFM model computes

$$\Phi_{\text{FmFM}}(\boldsymbol{x}) = w_0 + \langle \boldsymbol{x}, \boldsymbol{w} \rangle + \sum_{i=1}^{n} \sum_{j=i+1}^{n} \langle x_i \boldsymbol{v}_i, x_j \boldsymbol{v}_j \rangle_{\boldsymbol{M}_{f(i),f(j)}}, \tag{1}$$

where $w_0 \in \mathbb{R}$, $\boldsymbol{w} \in \mathbb{R}^n$, and $\boldsymbol{v}_i \in \mathbb{R}^{k_i}$ are learned parameters. The *field-interaction matrices* $\boldsymbol{M}_{f(i),f(j)} \in \mathbb{R}^{k_{f(i)} \times k_{f(j)}}$ can be either learned or predefined, and may have a special structure. Note that this model family allows a different embedding dimension for each field, since the matrices $\boldsymbol{M}_{f(i),f(j)}$ do not have to be square.

Let us explain why all the models described here are special cases of FmFMs in equation 1 in terms of their representation power: To recover the classical FMs, we take equation 1 with $\boldsymbol{M}_{f(i),f(j)} = \boldsymbol{I}$. To recover FFMs, let $\boldsymbol{P}_f$ be the matrix that extracts $\boldsymbol{v}_{i,f}$ from $\boldsymbol{v}_i$, namely, $\boldsymbol{P}_f \boldsymbol{v}_i = \boldsymbol{v}_{i,f}$. Then FFMs are also a special case of the FmFM model in equation 1 with $\boldsymbol{M}_{e,f} = \boldsymbol{P}_f^T \boldsymbol{P}_e$. Finally, to recover FwFMs we just need to take equation 1 with $\boldsymbol{M}_{e,f} = r_{e,f} \boldsymbol{I}$. Given this property, we use $\Phi_{\text{FmFM}}$ in equation 1 to describe and prove properties which should hold for the entire family.

## 2.4 Linearity of FmFMs and binning

A fundamental property of the FmFM family is that $\Phi_{\text{FmFM}}$ is *affine* in any one coordinate of $\boldsymbol{x}$. Formally, $\text{seg}[\Phi_{\text{FmFM}}, k](\boldsymbol{x}_{-k})$ is an affine function of $x_k$. The explanation stems from separating the summands with $x_k$ and those without $x_k$ in the $\Phi_{\text{FmFM}}$ formula:

$$\Phi_{\text{FmFM}}(\boldsymbol{x}) = \underbrace{w_0 + \langle \boldsymbol{x}_{-i}, \boldsymbol{w}_{-i} \rangle + \sum_{\substack{1 \leq i < j \leq n \\ i,j \neq k}} \langle x_i \boldsymbol{v}_i, x_j \boldsymbol{v}_j \rangle_{\boldsymbol{M}_{f(i),f(j)}}}_{\alpha} + x_k \underbrace{\left( w_k + \sum_{\substack{1 \leq j \leq n \\ j \neq k}} \langle \boldsymbol{v}_k, x_j \boldsymbol{v}_j \rangle_{\boldsymbol{M}_{f(i),f(j)}} \right)}_{\beta}$$

$$= \alpha + \beta x_k.$$

Consequently, an inherent limitation of the FmFM family is that it can represent only affine relationships between any one given input and the prediction. Thus, if the encoding function of a numerical field $f$ is linear scaling, such as min-max scaling or standardization, then $\Phi_{\text{FmFM}}^{\text{enc}} \equiv \Phi_{\text{FmFM}} \circ \mathbf{enc}$ will only be able to represent affine functions of that field. In most cases in practice, such an affine relationship is not expressive enough.

To overcome this limitation, numerical columns typically undergo *binning* or *quantization*: the numerical range is partitioned into a finite set of disjoint intervals, and $\mathbf{enc}_f(\cdot)$ one-hot encodes the interval its argument belongs to. Binning a numerical field $f$, this results in $\text{seg}[\Phi_{\text{FmFM}}^{\text{enc}}, f](\boldsymbol{z}_{-f})$ being a step function of $z_f$ whose jumps are at the binning boundaries. This is because the encoding of $z_f$ anywhere inside a given interval between two boundaries is constant, and thus the model's output is constant for any $z_k$ in that interval. Similarly, for two numerical fields $e < f$ the segmentized views $\text{seg}[\Phi_{\text{FmFM}}^{\text{enc}}, e, f](\boldsymbol{z}_{-(e,f)})$ are piecewise-constant functions of $(z_e, z_f)$, where the constant-value regions are grid cells formed by the bin boundaries of fields $e$ and $f$.

---

[1]FmFMs do not require it to be a true inner product. For it to be a true inner product, the matrix $\boldsymbol{P}$ has to be square and positive definite

## 2.5 Problem statement

When learning a model, our aim is for $\Phi_{\text{FmFM}}^{\text{enc}} = \Phi_{\text{FmFM}} \circ \mathbf{enc}$ to approximate some optimal predictor $\Phi^*$ as a function of the original tabular fields. Thus, we may think of segmentized views of $\Phi_{\text{FmFM}}^{\text{enc}}$ as approximations of the segmentized views of $\Phi^*$. Binning, therefore, results in piecewise-constant approximations. Binning with a very coarse grid may result in a poor approximation. Conversely, a grid that is too fine might significantly reduce the amount of training data falling into each interval, resulting in over-fitting. This kind of over-fitting is also known as the data sparsity issue. However, it is nothing more than the classical approximation-estimation balance.

With respect to this problem, our work has two aims - theoretical and practical. Theoretically, we aim to show that the interplay between a slightly modified FmFM model family and the classical well-known encoding schemes for numerical features, such as polynomials and splines, yields a powerful family of models: the segmentized views for each field are spanned by the chosen basis, whereas the views for each pair of fields are spanned by the basis tensor products, with the span coefficients depending on the remaining field. Consequently, the model learns a potentially different segmentized view for each segment, while the segmentized views inherit the approximation power of the chosen basis.

Second, we aim to utilize this theoretical insight to show how to improve an existing recommender system driven by models from the FmFM family trained with binned numerical features. Concretely, we show how this insight allows improving the approximation-estimation balance by avoiding sparsity for various kinds of numerical fields encountered in practice, and propose a deployment scheme for our modified models that requires only minor changes to an existing *serving*[2] part of a recommender system with only minor changes.

## 3 The basis function encoding approach

To describe our approach we need to introduce a slight modification to the FmFM computational graph formulated in equation 1. We can re-write the formula by extracting $x_i \boldsymbol{v}_i$ and $x_i w_i$ into auxiliary variables:

$$
\boldsymbol{U} = \begin{bmatrix} x_1 \boldsymbol{v}_1 \\ \vdots \\ x_n \boldsymbol{v}_n \end{bmatrix}
$$

$$
\boldsymbol{y} = \begin{pmatrix} x_1 w_1 \\ \vdots \\ x_n w_n \end{pmatrix}
$$

$$
\Phi_{\text{FmFM}} = w_0 + \sum_{i=1}^{n} y_i + \sum_{i=1}^{n} \sum_{j=i+1}^{n} \langle \boldsymbol{U}_{i,:}, \boldsymbol{U}_{j,:} \rangle \boldsymbol{M}_{f(i),f(j)}.
$$

Here, $\boldsymbol{U}_{i,:}$ is a "Python" notation for denoting the $i$-th row of $\boldsymbol{U}$. In a sense, the first two lines extract the relevant embedding vectors and linear coefficients, and the last line combines them to produce a score.

Our modification is applied to the first two 'extraction' lines. Before combining the vectors and the linear coefficients, we first apply a field-wise reduction to the embedding vectors and linear coefficients of each field. We consider two kinds of reductions: (a) the identity reduction, which just returns its input; and (b) a sum reduction, which sums up the embedding vectors belonging to the field: $\sum_{i \in \mathcal{I}(f)} x_i \boldsymbol{v}_i$ and, respectively, $\sum_{i \in \mathcal{I}(f)} x_i w_i$, where $\mathcal{I}(f)$ are the indices in $\boldsymbol{x}$ containing the encoding of field $f$.

The identity reduction is equivalent to multiplying by the identity matrix on the left, whereas the summation reduction is equivalent to multiplying by a single-row matrix of ones $\mathbf{1}_\ell^T$ of the appropriate size $\ell$. Thus, each field $f$ has an associated reduction matrix $\boldsymbol{R}_f$, and the resulting model is obtained by pre-multiplying the original $\boldsymbol{U}$ and $\boldsymbol{y}$ by a block-diagonal reduction matrix that applies the appropriate reduction the embedding

---

[2]The serving part of a recommender system is the part that consumes trained models, and uses them to recommend items.

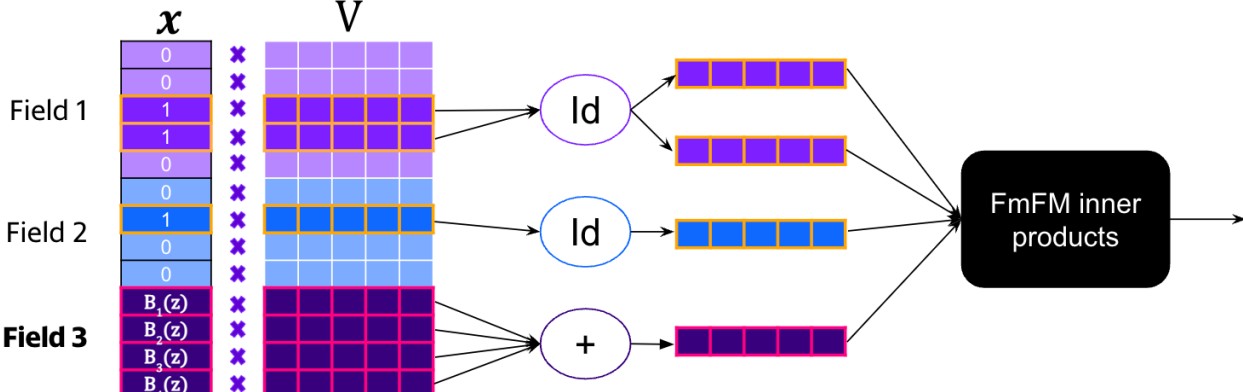

Figure 2: The computational graph with continuous numerical fields. Field 3 is a continuous numerical field whose value is $z$. The vectors $\boldsymbol{v}_1, \ldots, \boldsymbol{v}_n$ in the rows of $\boldsymbol{v}$ are multiplied by the input vector $\boldsymbol{x}$. Then, a reduction is applied to each field. Most fields have the identity (Id) reduction, whose output is identical to its input, whereas field 3 uses the sum reduction. The resulting vectors are then passed to the pairwise interaction module. An analogous process happens with the $\boldsymbol{w}$ vector.

vectors / linear weights belonging to each field:

$$
\boldsymbol{U} = \begin{bmatrix} \boldsymbol{R}_1 & & \\ & \ddots & \\ & & \boldsymbol{R}_m \end{bmatrix} \begin{bmatrix} x_1 \boldsymbol{v}_1 \\ \vdots \\ x_n \boldsymbol{v}_n \end{bmatrix}
$$

$$
\boldsymbol{y} = \begin{bmatrix} \boldsymbol{R}_1 & & \\ & \ddots & \\ & & \boldsymbol{R}_m \end{bmatrix} \begin{pmatrix} x_1 w_1 \\ \vdots \\ x_n w_n \end{pmatrix} \tag{2}
$$

$$
\Phi_{\mathrm{FmFM}} = w_0 + \sum_{i=1}^{\hat{n}} y_i + \sum_{i=1}^{\hat{n}} \sum_{j=i+1}^{\hat{n}} \langle \boldsymbol{U}_{i,:}, \boldsymbol{U}_{j,:} \rangle_{\boldsymbol{M}_{f(i),f(j)}}.
$$

For every field $f$, the matrix $\boldsymbol{R}_f$ is either the identity matrix, or a row of ones. Note, that after the reductions, the matrix $\boldsymbol{U}$ and the vector $\boldsymbol{y}$ may have a different number of rows, denoted by $\hat{n}$ in the formula above.

As a slight abuse of notation, we shall now refer to this slightly modified model in equation 2 as "the" FmFM. This is because an FmFM is obtained by choosing all reductions to be the identity. Next, we shall describe how we incorporate basis encoding into this model.

Suppose we would like to encode a continuous numerical field $f$. We choose a set of functions $B_1^f, \ldots, B_{\ell_f}^f$, and encode the field as $\mathbf{enc}_f(z) = (B_1^f(z), \ldots, B_{\ell_f}^f(z))^T$. For that field, we choose the summing reduction, namely, $\boldsymbol{R}_f$ is a row of ones. When the field $f$ under consideration if clear from the context, we omit it for clarity, and write $B_1, \ldots, B_\ell$. The result is that for this field, there is *one* row in $\boldsymbol{U}$ that is equal to $\sum_{i \in \mathcal{I}(f)} B_i(z) \boldsymbol{v}_i$ and one row in $\boldsymbol{y}$ that is equal to $\sum_{i \in \mathcal{I}(f)} B_i(z) w_i$. Note that each field can have its own set of basis functions, and in particular, the size of the bases may differ. The computational graph, including basis encoding, is depicted in Figure 2. We note that the block-diagonal matrix is only a mathematical formalism, and a reasonable implementation to compute the score is similar to what is illustrated in the figure: extract vectors corresponding *only* to the non-zero components of $\boldsymbol{x}$ associated with each field, perform the reductions for each field, and then compute the pairwise inner products.

As a final note, observe that binning is obtained as a special case of our basis encoding method. Indeed, suppose we have $\ell$ intervals that partition our numerical range, choosing the basis $B_1, \ldots, B_\ell$ to be just the indicator functions of these intervals yields a model $\Phi_{\mathrm{FmFM}}^{\mathrm{enc}}$ that is equivalent to binning.

### 3.1 Spanning properties

To show why our modeling choices improve the approximation power of the model, we present two technical lemmas showing that the segmentized functions for one or two basis-encoded tabular columns are spanned by the basis of choice, or the tensor product of the two chosen bases.

**Lemma 1** (Spanning property). *Let*

$$\mathbf{enc}_f(z) = (B_1(z), \cdots, B_\ell(z))^T$$

*be the encoding associated with a* continuous numerical field $f$, *let* $\boldsymbol{R}_f = \mathbf{1}_\ell^{T3}$, *and suppose* $\Phi_{\mathrm{FmFM}}$ *is computed according to equation 2. Then, for any* $\boldsymbol{z}_{-f}$ *there exist* $\alpha_1, \ldots, \alpha_\ell, \beta \in \mathbb{R}$, *which depend only on* $\boldsymbol{z}_{-f}$, *such that:*

$$\mathrm{seg}[\Phi_{\mathrm{FmFM}}^{\mathrm{enc}}, f](\boldsymbol{z}_{-f})(z) = \sum_{i=1}^{\ell} \alpha_i B_i(z) + \beta.$$

An elementary formal proof can be found in Appendix A.1, but it can be explained intuitively. Looking at equation 2, both the matrix $\boldsymbol{U}$ and the vector $\boldsymbol{y}$ have *one* row that is linear in the basis $B_1, \ldots, B_\ell$, whereas the other rows are constant. Since $\Phi_{\mathrm{FmFM}}$ is affine in any *one* input feature, it must be an affine function of the basis.

It becomes even more interesting when looking at segmentized functions for *two* continuous numerical fields.

**Lemma 2** (Pairwise spanning property). *Let* $e < f$ *be two continuous numerical fields. Let*

$$\mathbf{enc}_e(z_e) = (B_1(z_e), \cdots B_\ell(z_e))^T, \quad enc_f(z_f) = (C_1(z_f), \cdots C_\kappa(z_f))^T,$$

*let* $\boldsymbol{R}_e = \mathbf{1}_\ell^T$ *and* $\boldsymbol{R}_f = \mathbf{1}_\kappa^T$, *and suppose* $\Phi_{\mathrm{FmFM}}$ *is computed according to equation 2. Define* $B_0(z) = C_0(z) = 1$. *Then, for every* $\boldsymbol{z}_{-(e,f)}$, *there exist* $\alpha_{i,j}, \beta \in \mathbb{R}$ *for* $i \in \{0, \ldots, \ell\}$ *and* $j \in \{0, \ldots, \kappa\}$, *such that:*

$$\mathrm{seg}[\Phi_{\mathrm{FmFM}}^{\mathrm{enc}}, e, f](\boldsymbol{z}_{-(e,f)})(z_e, z_f) = \sum_{i=0}^{\ell} \sum_{j=0}^{\kappa} \alpha_{i,j} B_i(z_e) C_j(z_f) + \beta.$$

The proof can be found in Appendix A.2. The intuition is similar in nature to the one we have for the spanning property for one field, but now *two* instead of one rows of $\boldsymbol{y}$ and $\boldsymbol{U}$ are affine in the bases.

It's important to note that the segmentized view $\mathrm{seg}[\Phi_{\mathrm{FmFM}}^{\mathrm{enc}}, e, f](\boldsymbol{z}_{-(e,f)})$ has $O(\ell \cdot \kappa)$, but the model does not actually learn $O(\ell \cdot \kappa)$ parameters. Instead, it learns $O(\ell + \kappa)$ parameters in the form of $\ell + \kappa$ embedding vectors. Consequently, the spanning coefficient matrix $\boldsymbol{\alpha} = (\alpha_{i,j})_{i,j=0}^{\ell,\kappa}$ for each segment is not learned explicitly, but rather its low-rank factorization is learned in the form of embedding vectors. This extends the observation made in Rügamer (2022) for classical FMs to the broader FmFM family.

### 3.2 Splines and the B-Spline basis

Spline functions (Schoenberg, 1946) are piece-wise polynomial functions of degree $d$ with up to $d-1$ continuous derivatives defined on some interval $[a, b]$. The interval is divided into disjoint sub-intervals at a set of break-points $a = t_0 < t_1 < \cdots < t_{\ell-d} = b$, where each polynomial piece is defined on $[t_j, t_{j+1}]$. It is well-known that spline functions of degree $d$ defined on $\ell - d + 1$ break-points can be written as weighted sums of the celebrated B-Spline basis (de Boor, 2001, pp 87) comprising of exactly $\ell$ functions. For brevity, we will not elaborate their explicit formula in this paper, and point out that it's available in a variety of standard scientific computing packages, e.g., the `scipy.interpolate.BSpline` class of the SciPy package (Virtanen et al., 2020).

In this paper we concentrate on the cases of $d = 2$ and $d = 3$, which are *quadratic* and *cubic splines*, respectively, with uniformly spaced break-points. At this stage we assume that the values of our numerical field lie in a compact interval $[a, b]$, which we assume w.l.o.g is $[0, 1]$. We discuss the more generic cases in the

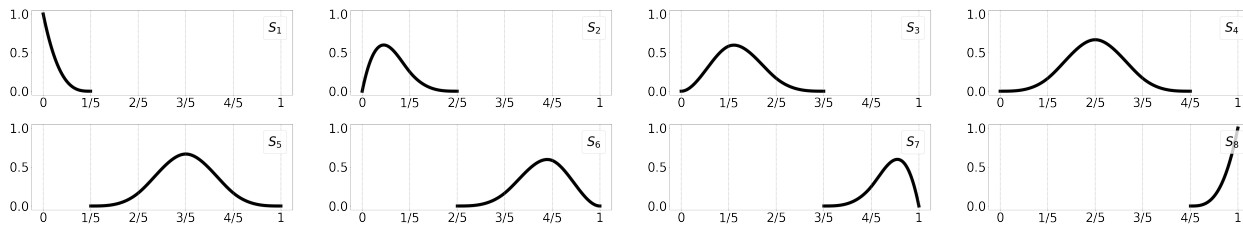

Figure 3: Cubic B-Spline basis functions defined on five break-points.

following sub-section. A visual depiction of the cubic B-Spline basis with five break-points is illustrated in Figure 3.

The B-Spline basis is our proposed candidate for recommender systems for two reasons: computational efficiency, and approximation power. For efficiency, an important property of the B-Spline basis of degree $d$ is that at any point only $1 + d$ basis functions are non-zero. Thus, regardless of the number of basis functions $\ell$ we use, computing the model's output remains efficient, since the reduction described in Figure 2 requires a weighted sum of only $1 + d$ vectors, regardless of the size of the basis.

For the approximation power, it is known (de Boor, 2001, pp 149), that splines of degree $d$ can approximate an arbitrary function $g$ with $k \leq 1 + d$ continuous derivatives up to an error bounded by $O(\|g^{(k)}\|_\infty / \ell^k)^4$, where $g^{(k)}$ is the $k^{\text{th}}$ derivative of $g$. The spanning property (Lemma 1) ensures that the model's segmentized outputs are splines spanned by the same basis, and therefore are more powerful in approximating the optimal segmentized outputs than step functions. Assuming that the functions we aim to approximate are smooth enough and vary "slowly", in the sense that their high-order $k^{\text{th}}$ derivatives are small, the approximation error goes down at the rate of $O(\frac{1}{\ell^k})$, whereas with binning the rate is $O(\frac{1}{\ell})$.

A direct consequence is that we can obtain a theoretically good approximation which is also achievable in practice, since we can be accurate with a small number of basis functions, and this significantly decreases the chances of sparsity and over-fitting issues. This is in contrast to binning, where high-resolution binning is required to for a good theoretical approximation accuracy, but it may not be achievable in practice.

### 3.3 Continuous numerical fields with arbitrary domain and distribution

Splines approximate functions on a compact interval. Thus, numerical fields with unbounded domains pose a challenge. Moreover, the support of each B-Spline function is only a sub-interval of the domain defined by $1 + d$ consecutive knots. Thus, even if a numerical field $f$ is bounded in $[a, b]$, a highly skewed distribution may cause "starvation" of the support of some basis functions: if $\mathbb{P}(z_f \in \text{support}(B_i))$ is extremely small, there will be little training data to effectively learn a useful representation of $B_i$.

Here we do not invent any wheels, and suggest using common machine-learning practice: a scalar transform $T_f : \mathbb{R} \to [0, 1]$ to the features of field $f$, such as min-max scaling, or quantile transform[5]. We note that standardization is less appropriate, since standardized values do not necessarily lie in a bounded interval, as required by the B-Spline basis.

The quantile transform idea is useful for conducting benchmarks, but in a real-time recommender system it may be prohibitive due to its computational complexity. A quantile transform is obtained by learning the empirical CDF of the training data, and using it as the transform $T_f$. But the empirical CDF is just an approximation of the data generating distribution, and we may use others. Simple parametric distributions have closed-form and fast to compute CDF formulae, and we found that in practice it works well: just fit a few parametric distributions to the training data of a given field $f$, and use the best-fit as $T_f$. We found this to work well in practice, as shown in Section 4.4.

---

[3]$\mathbf{1}_\ell^T$ is a *row* vector of $\ell$ ones
[4]For a function $\phi$ defined on $S$, its infinity norm is $\|\phi\|_\infty = \max_{x \in S} |\phi(x)|$
[5]Replacing $z_f$ with eCDF($z_f$), where eCDF is the empirical CDF learned from the training data

Over-all, the property we want is for $\mathbb{P}(z_f \in \text{support}(B_i))$ to non-negligible for every basis function $B_i$, namely, the data is spread on the interval such that no region of the interval is "starved". Thus, even though that in the experiments section we use min-max scaling and quantile transform, for a practical large-scale recommender system we recommend using a parametric distribution.

The above discussion shows that our method does not eliminate the need for data-analysis and feature engineering, but only streamlines it: just try out a few simple options to make sure the data is not too skewed when mapped to $[0, 1]$. The rest of the "heavy lifting" is done by the strong approximation power splines, and the computational efficiency facilitated by the B-Spline basis.

### 3.4  Integration into an existing system by simulating binning

Suppose we would like to obtain a model which employs binning of the field $f$ into a large number $N$ of intervals, e.g. $N = 1000$. As we discussed, in most cases we cannot directly learn such a model because of the approximation-estimation balance. However, we can *generate* such a model from another model trained using our scheme to make initial integration easier.

The idea is best explained by referring, again, to Figure 2 and equation 2. For a given numerical field $f$, the reduction stage produces only *one* row of of the post-reduction matrix $\boldsymbol{U}$, which we shall denote by $\boldsymbol{u}$. In fact, this row is a function of $z_f$, namely:

$$\boldsymbol{u}(z_f) = \sum_{i \in \mathcal{I}(f)} \boldsymbol{v}_i B_i(T_f(z_f)).$$

Consequently, we have a mapping from $z_f$ to a corresponding embedding vector. It is a curve in the $k$-dimensional space, parametrized by $z_f$. Hence, to simulate $N$ "bins" $([z_{j+1}, z_j))_{j=0}^N$, we simply discretize this curve at $N$ points, such as the mid-point of each interval:

$$\boldsymbol{u}(\tfrac{z_{j+1}+z_j}{2}) \quad j = 0, \dots, N$$

The resulting model now has an embedding vector for each bin, as we desire. Hence, the serving system sees a binning-based model with many bins, even though it was trained with an arbitrary basis of our choice.

## 4  Evaluation

We divide this section into three parts. First, we use a synthetically generated data-set to show that our theory holds - the model learns segmentized output functions that resemble the ground truth. Then, we compare the accuracy obtained with binning versus splines on several data-sets. The code to reproduce these experiments is available in the supplemental material. Finally, we report the results of a web-scale A/B test conducted on a major online advertising platform serving real traffic.

### 4.1  Learning artificially chosen functions

We used a synthetic toy  click-through rate prediction data-set with four fields, and zero-one labels (click / non-click). Naturally, the cross-entropy loss is used to train models on such tasks. We have three categorical fields each having two values each, and one numerical field in the range $[0, 40]$. For each of the eight segment configurations defined by the categorical fields, we defined functions $p_0, \dots, p_7$ (see Figure 4) describing the CTR  as a function of the numerical field. Then, we generated a data-set of 25,000 rows, such that for each row $i$ we chose a segment configuration $s_i \in \{0, ..., 7\}$ of the categorical fields uniformly at random, the value of the numerical field $z_i \sim \text{Beta-Binomial}(40, 0.9, 1.2)$, and a label $y_i \sim \text{Bernoulli}(p_{s_i}(z_i))$.

We trained an FFM (Juan et al., 2016) provided by Yahoo-Inc (2023) using the binary cross-entropy loss on the above data, both with binning of several resolutions and with splines defined on 6 sub-intervals . The numerical field was naïvely transformed to $[0, 1]$ by simple normalization. We plotted the learned curve for every configuration in Figure 4. Indeed, low-resolution binning approximates poorly, a higher resolution approximates better, and a too-high resolution cannot be learned because of sparsity. However, Splines defined on only six sub-intervals approximate the synthetic functions $\{p_i\}_{i=0}^7$ quite well.

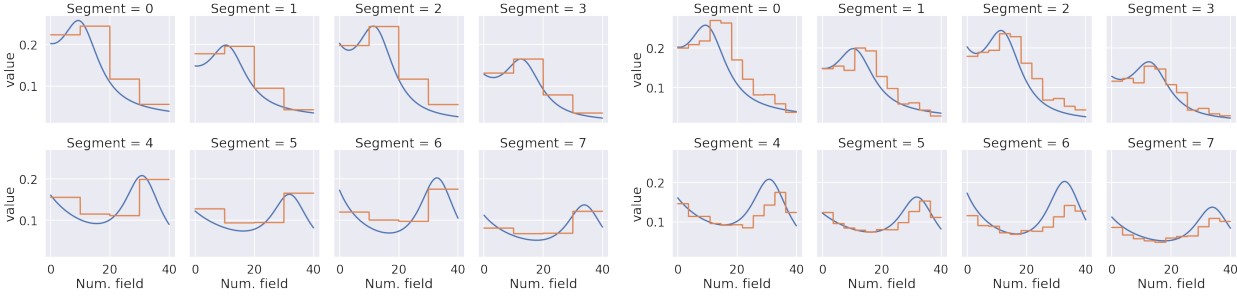

(a) 5-bin segmentized approximation (test loss = 0.3474)   (b) 12-bin segmentized approximation (test loss = 0.3442)

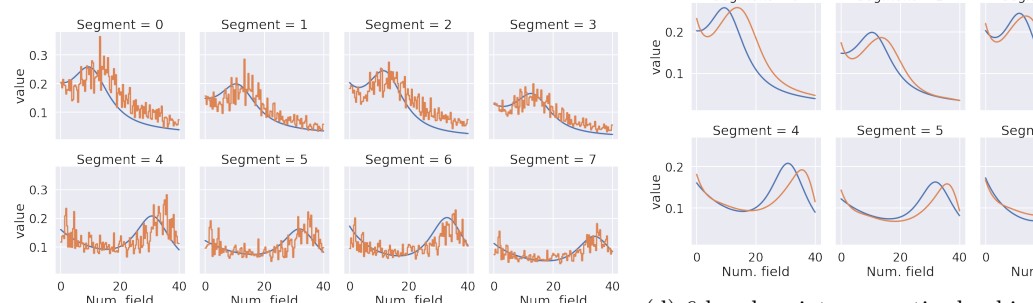

(c) 120-bin segmentized approximation (test loss = 0.3478)

(d) 6 break-point segmentized cubic spline approximation (0.3432)

Figure 4: Results of segmentized approximations of four FFM models trained on synthetic data. In each plot, a family of segmentized functions on the interval $[0, 40]$, plotted in blue, are approximated by the model, plotted in orange. 12 bins are more accurate than 5, but 120 bins are even less accurate than 5 due to sparsity. With splines we achieve best accuracy.

Next, we compared the test cross-entropy loss on 75,000 samples generated in the same manner with for several numbers of intervals used for binning and cubic Splines. For each number of intervals we performed 15 experiments to neutralize the effect of random model initialization. As is apparent in Figure 5, Splines consistently outperform in this theoretical setting. The test loss obtained by both strategies increases if the number of intervals becomes too large, but the effect is much more significant in the binning solution.

## 4.2 Public tabular data-sets

Since our approach works on any tabular dataset, and isn't specific to recommender systems, we mainly test our approach versus binning on several tabular data-sets with abundant numerical features that have a strong predictive power: the California housing (Pace & Barry, 1997) , adult income (Kohavi, 1996), Higgs (Baldi et al., 2014) (we use the 98K version from OpenML (Vanschoren et al., 2014)), and song year prediction (Bertin-Mahieux et al., 2011). For the first two data-sets we used an FFM, whereas for the last two we used an FM, both provided by Yahoo (Yahoo-Inc, 2023), since FFMs are significantly more expensive to train when there are many columns, and even more so with hyper-parameter tuning. The above data-sets were chosen since they were used in a previous line of work by Gorishniy et al. (2021; 2022) on numerical features in tabular data-sets. We chose the subset of these data-sets whose labels are either real-valued or binary, since multi-class or multi-label classification problems require a model that produce a scalar for each class. Factorization machines are limited to only one scalar in their output.

We assume that the task on all data-sets is regression, both with real-valued and binary labels. This is in line with what factorization machines are commonly used for - CTR prediction. For binary labels we use the cross-entropy loss, whereas for real-valued labels we use the L2 loss. For tuning the step-size, batch-size, the number of intervals, and the embedding dimension we use Optuna (Akiba et al., 2019). For binning, we also tuned the choice of uniform or quantile bins. In addition, 20% of the data was held out for validation, and

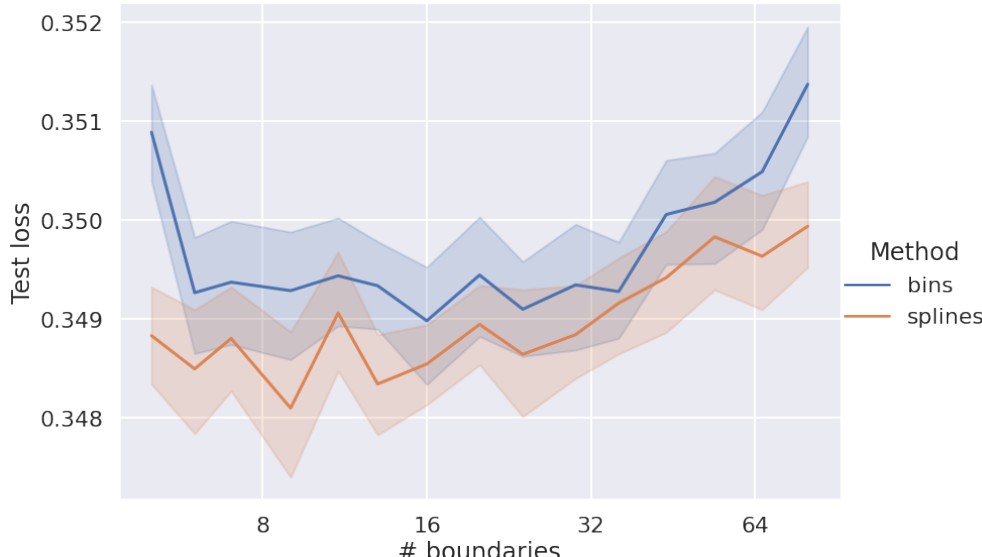

Figure 5: Comparison of the test cross-entropy loss obtained with Splines and bins. Both methods suffer from sparsity issues as the number of intervals grows, but Splines are able to utilize their approximation power with a small number of intervals, before sparsity takes effect. The bands are 90% bootstrap confidence intervals based on multiple experiment repetitions: for each number of boundaries and numerical field type we ran 15 experiments to neutralize the effect of random initialization.

regression targets were standardized. Finally, for the adult income data-set, 0 has a special meaning for two columns, and was treated as a categorical value.

We ran 20 experiments with the tuned configurations to neutralize the effect of random initialization, and report the mean and standard deviation of the metrics on the test set in Table 1, where it is apparent that our approach outperforms binning on these datasets. These datasets were chosen since they contain several numerical fields, and are small enough to run many experiments to neutralize the effect of hyper-parameter choice and random initialization at a reasonable computational cost, or time. They were also used in other works on tabular data, such as Gorishniy et al. (2021; 2022).

Table 1: Comparison of binning vs. splines. Standard deviations are reported as in parentheses as % of the mean.

|  | Cal. housing | Adult income | Higgs | Song year |
| --- | --- | --- | --- | --- |
| # rows | 20640 | 48842 | 98050 | 515345 |
| # cat. fields | 0 | 8 | 0 | 0 |
| # num. fields | 8 | 6 | 28 | 88 |
| Metric type | RMSE | Cross-Entropy | Cross-Entropy | RMSE |
| Binning metric (std%) | 0.4730 (0.36%) | 0.2990 (0.47%) | 0.5637 (0.22%) | 0.9186 (0.4%) |
| Splines metric (std%) | **0.4294** (0.57%) | **0.2861** (0.28%) | **0.5448** (0.12%) | **0.8803** (0.2%) |
| Splines vs. binning lift | 9.2% | 4.3% | 3.36% | 4.16% |

## 4.3 The Criteo dataset

To further demonstrate the benefits of our approach over on a real-world recommendation dataset, we evaluate it with an FwFM on the Criteo display advertising challenge dataset (Criteo, 2014). It has plenty of numerical fields, but these fields are *integers*. Before presenting the experiments, it's important to discuss the unique

challenges posed by integers, and especially fields representing counts, e.g. the number of visits of the user in the last week.

We note here that modeling integers is *not* in the scope of this work, but a paper on recommender systems cannot go without experiments with large scale recommender-system datasets. The Criteo dataset was chosen due to its popularity and the abundance of numerical fields, even though they are integers. We point out that other classical public recommendation datasets, such as MovieLens (GorupLens), or Avazu (Avazu, 2014), do not contain continuous or similar integer numerical fields, and therefore were incompatible for the analysis of this work[6] Thus, what we propose doing with integers in this section is more of a heuristic adaptation of our method to enable conducting an experiment with the Criteo data-set, rather than systematic treatment of integers that can be seen as an integral part of this paper.

Integers possess properties of both discrete and continuous nature. For example, their CDF function is a *step* function that may have large jumps, and it's not trivial to transform them to $[0, 1]$ using a CDF approximation in a reasonable manner. This is because large jumps produce large gaps in the $[0, 1]$ interval that are not covered by any data. Moreover, features that represent counts, such as the number of times the user interacted with some category of items, pose an additional difficulty stemming from this hybrid discrete-continuous nature. Smaller values are more 'discrete', while larger values are more 'continuous', i.e. the difference between users who never visited our site and users who visited it once may be large, the difference between one and two visits will be large as well, but the difference in user behavior between 98 and 99 visits is probably small.

The data-set is a log of 7 days of ad impressions and clicks in chronological order. Thus, we split into training, validation, and test in the following manner: the first $5/7$ of the data-set is the training set, the next $1/7$ is the validation set for hyper-parameter tuning, and the last $1/7$ is the test set. Categorical features with less than 10 occurrences were replaced by a special "INFREQUENT" value for every field . When employing binning, we use a similar strategy to the winners of the Criteo challenge - the bin index is $\text{floor}(\ln^2(z))$ for $z \geq 1$. Namely, the bin boundaries are $\{\exp(\sqrt{i})\}_{i=0}^{\ell}$ for $z \geq 1$. Values smaller than 1 are treated as categorical values. The lower and upper bounds are learned from the training set.

For splines, we stand on the shoulders of giants, and use a transform that resembles the squared logarithm $x \to \text{arcsinh}^2(x)$, followed min-max scaling. The reason is that $\text{arcsinh}(x)$ mimics the logarithm, but is also defined for zero. The min-max scaling ensures that the transformed feature stays in $[0, 1]$, and is learned from the training set only. If the validation or the test set contain values above the maximum observed, they are mapped to 1. Negative values are, similarly, treated as categorical values. We note, that this transform is far from being the empirical CDF, and due to the reasons discussed above, the empirical CDF is typically not applicable to integers. The histograms of the transformed columns `I_1`, ..., `I_13` are plotted in Figure 6. We can see that some of the columns appear to be quite discrete. For example, `I_10` has only *four* distinct values. The columns `I_11` and `I_12` appear discrete as well. Thus, when conducting an experiment with cubic splines, we used them for all columns, except for the above three.

We conduct experiments with $k \in 8, 16, \ldots, 64$ as embedding dimensions, and each experiment is conducted using 50 trials of Optuna (Akiba et al., 2019) with its default configuration to tune the learning rate and the $L_2$ regularization coefficient. The models were trained using the AdamW optimizer (Loshchilov & Hutter, 2019). As an ablation study, to make sure that cubic splines contribute to the the improvement we observe, we also conduct experiments with $0^{th}$ order splines applied after the above transformation, since it may be the case that the arcsinh transformation itself yields an improvement. We remind the readers that $0^{th}$ order splines are just uniform bins. For splines, we used 20 knots, which is roughly half the number of bins obtained by the strategy employed by the Criteo winners. The obtained test losses are summarized in Figure 7. It is apparent that cubic splines outperform both uniform binning ($0^{th}$ order splines) and the original binning procedure of the Criteo winners for most embedding dimensions. Moreover, we can see that cubic splines perform best when the embedding dimension is slightly larger than the best one for binning. We conjecture that, at least for the Criteo data-set, cubic splines require more expressive power yielded by a slightly higher embedding dimension to show their full potential.

---

[6]We could, in theory, engineer such features using the well-known Target Encoding or Count Encoding techniques, but this is out of the scope of this work.

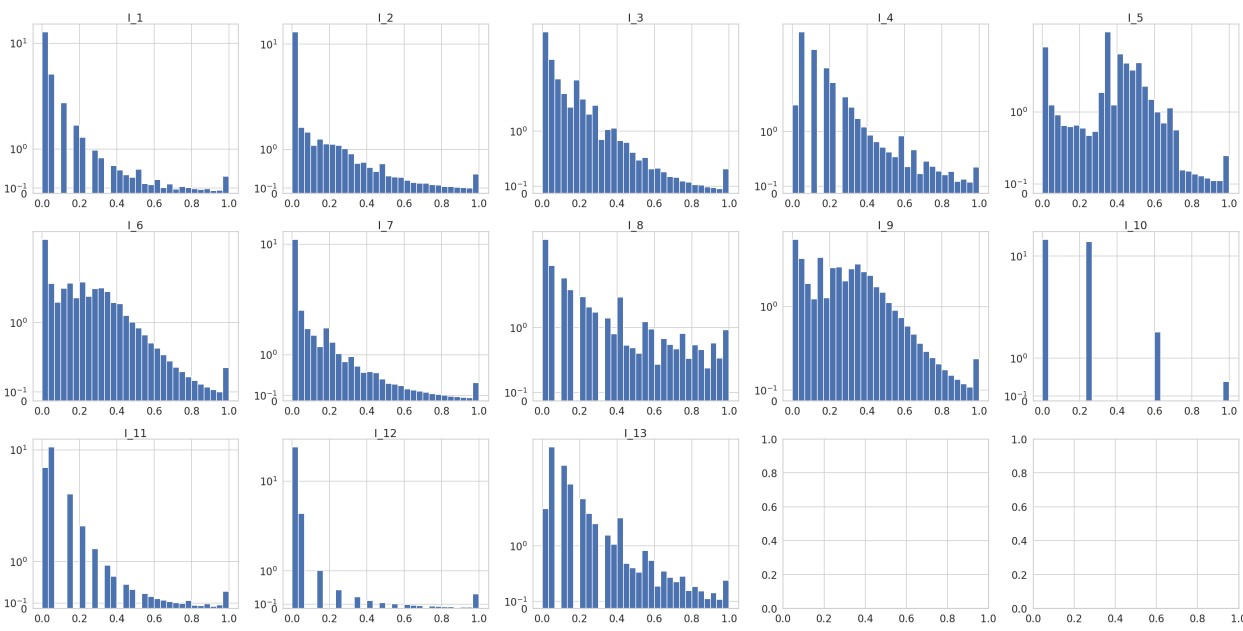

Figure 6: Histograms of columns I_1, ..., I_13 transformed by $T_f(x)$ defined by the squared inverse hyperbolic sine, followed by min-max scaling.

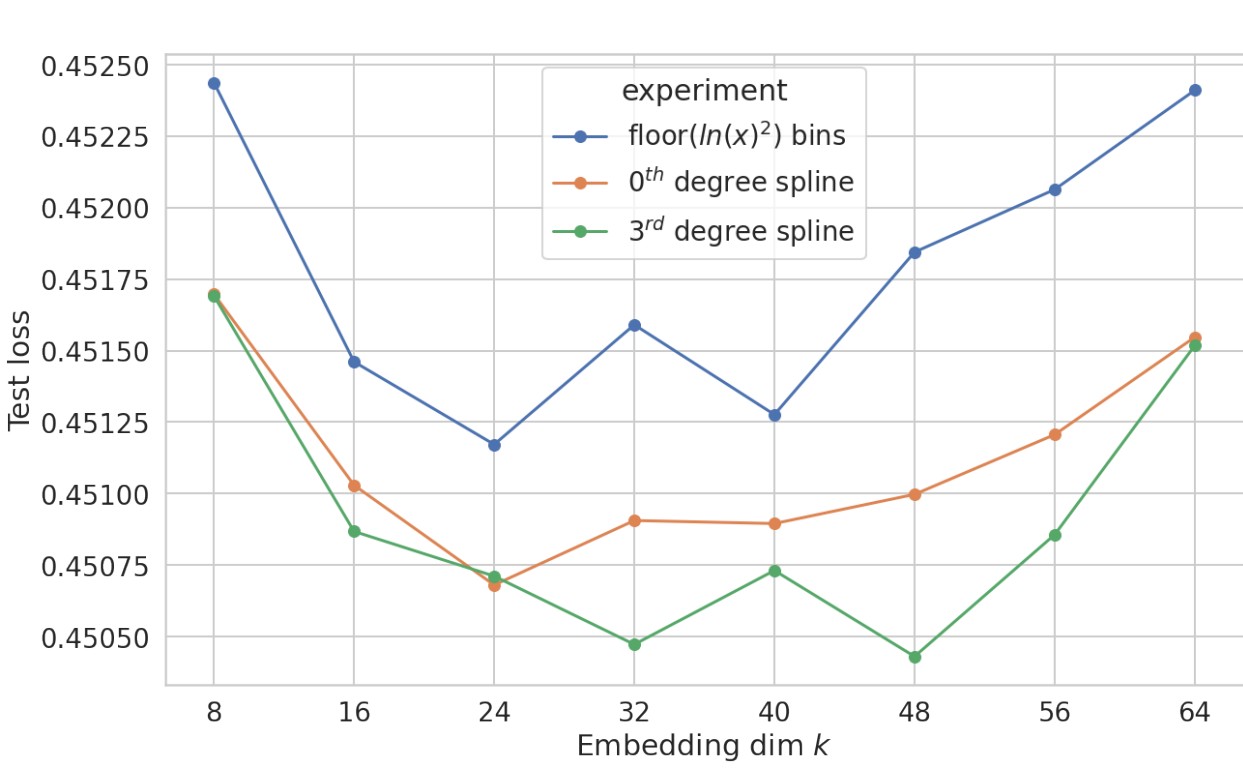

Figure 7: Test LogLoss for the Criteo experiment for various embedding dimensions.

### 4.4 A/B test results on an online advertising system

Here we report an online performance improvement measured using an A/B test, serving real traffic of a major online advertising platform. The platform applies a proprietary CTR prediction model that is closely related to FwFM (Sun et al., 2021). The model, which provides CTR predictions for merchant catalogue-based ads, has a *recency* field that measures the time (in hours) passed since the user viewed a product at the advertiser's site. We compared an implementation using our approach of continuous feature training and high-resolution binning during serving time described in Section 3.4 with a fine grained geometric progression of 200 bin break points, versus the "conventional" binned training and serving approach used in the production model at that time. The new model is only one of the rankers[7] that participates in our ad auction. Therefore, a mis-prediction means the ad either unjustifiably wins or loses the auction, both leading to revenue losses.

We conducted an A/B test against the production model at that time, when our new model was serving 40% of the traffic for over six days. The new model dramatically reduced the CTR prediction error, measured as $\left(\frac{\text{Average predicted CTR}}{\text{Measured CTR}} - 1\right)$ on an hourly basis, from an average of 21% in the baseline model, to an average of 8% in the new model. The significant increase in accuracy has resulted in this model being adopted as the new production model.

## 5 Discussion

We presented an easy to implement approach for improving the accuracy of the factorization machine family whose input includes numerical fields. Our scheme avoids increasing the number of model parameters and over-fitting, by relying on the approximation power of splines. This is explained by the spanning property along with the spline approximation theorems. Moreover, the discretization strategy described in Section 3.4 allows our idea to be integrated into an existing recommendation system without introducing major changes to the production code that utilizes the model to rank items, i.e., inference. .

It is easy to verify that our idea can be extended to factorization machine models of higher order (Blondel et al., 2016). This has been shown in Rügamer (2022) for vanilla higher order FMs, but can be extended to "field-informed" variants of higher order FMs. In particular, the spanning property in Lemma 1 still holds, and the pairwise spanning property in Lemma 2 becomes $q$-wise spanning property from machines of order $q$. However, to keep the paper focused and readable, we keep the analysis out of the scope of this paper.

With many advantages, our approach is not without limitations. We do not eliminate the need for data research and feature engineering, which is often required when working with tabular data, since the data still needs to be analyzed to fit a function that roughly resembles the empirical CDF. We believe that feature engineering becomes easier and more systematic, but some work still has to be done.

Finally, we would like to note two drawbacks. First, our approach slightly reduces interpretability, since we cannot associate a feature with a corresponding learned latent vector. Second, our approach may not be applicable to all kinds of numerical fields. For example, consider a product recommendation system with a product price field. Usually higher prices mean a different category of products, leading to a possibly different trend of user preferences. In that case, the optimal segmentized output as a function of the product's price is probably far from having small (higher order) derivatives, and thus cubic splines may perform poorly, and possibly even worse than simple binning.

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

# A  Proof of the spanning properties

## A.1  Proof of the spanning property (Lemma 1)

*Proof.* For some vector $\mathbf{q}$, denote by $\mathbf{q}_{a:b}$ the sub-vector $(q_a, \ldots, q_b)$. Assume w.l.o.g. that $f = 1$, and that field 1 has the value $z$. By construction in equation 2, we have $y_1 = \sum_{i=1}^{\ell} w_i B_i(z)$, while the remaining components $\boldsymbol{y}_{2:\hat{n}}$ do not depend on $z$. Thus, the we have

$$w_0 + \langle \mathbf{1}, \boldsymbol{y} \rangle = \sum_{i=1}^{\ell} w_i B_i(z) + \underbrace{w_0 + \langle \mathbf{1}, \hat{\boldsymbol{y}}_{2:\hat{n}} \rangle}_{\beta_1}. \tag{3}$$

Moreover, by equation 2 we have that $\boldsymbol{U}_{1,:} = \sum_{i=1}^{\ell} \boldsymbol{v}_i B_i(z)$, whereas the remaining rows $\boldsymbol{U}_{2,:}, \ldots, \boldsymbol{U}_{\hat{n},:}$ do not depend on $z$. Thus,

$$
\begin{aligned}
\sum_{i=1}^{\hat{n}} \sum_{j=i+1}^{\hat{n}} & \langle \boldsymbol{U}_{i,:}, \boldsymbol{U}_{j,:} \rangle M_{f(i),f(j)} \\
&= \sum_{j=2}^{\hat{n}} \langle \boldsymbol{U}_{1,:}, \boldsymbol{U}_{j,:} \rangle M_{1,f(j)} + \underbrace{\sum_{i=2}^{\hat{n}} \sum_{j=i+1}^{\hat{n}} \langle \boldsymbol{U}_{i,:}, \boldsymbol{U}_{j,:} \rangle M_{f(i),f(j)}}_{\beta_2} \\
&= \sum_{j=2}^{\hat{n}} \langle \sum_{i=1}^{\ell} \boldsymbol{v}_i B_i(z), \boldsymbol{U}_{j,:} \rangle M_{1,f(j)} + \beta_2 \\
&= \sum_{i=1}^{\ell} \underbrace{\left( \sum_{j=2}^{\hat{n}} \langle \boldsymbol{v}_i, \boldsymbol{U}_{j,:} \rangle M_{1,f(j)} \right)}_{\tilde{\alpha}_i} B_i(z) + \beta_2.
\end{aligned}
\tag{4}
$$

Combining equation 3 and equation 4, we obtain

$$\mathrm{seg}[\Phi_{\mathrm{FmFM}}^{\mathrm{enc}}, 1](\boldsymbol{z}_{-1})(z_1) = \sum_{i=1}^{\ell} (w_i + \tilde{\alpha}_i) B_i(z) + (\beta_1 + \beta_2),$$

which is of the desired form. $\qquad\square$

## A.2  Proof of the pairwise spanning property (Lemma 2)

*Proof.* Recall that we need to rewrite equation 2 as a function of $z_e, z_f$, which are the values of the fields $e$ and $f$. Assume w.l.o.g. that $e = 1, f = 2$. By construction in equation 2, we have $y_1 = \sum_{i=1}^{\ell} w_i B_i(z_1)$, and $y_2 = \sum_{i=1}^{\kappa} w_i C_i(z_2)$, while the remaining components $\hat{\boldsymbol{y}}_{3:\hat{n}}$ do not depend on $z$. Thus, the we have

$$
\begin{aligned}
w_0 + \langle \mathbf{1}, \boldsymbol{y} \rangle &= \sum_{i=1}^{\ell} w_i B_i(z_1) + \sum_{i=1}^{\kappa} w_{i+\ell} C_i(z_2) + \underbrace{w_0 + \langle \mathbf{1}, \boldsymbol{y}_{3:\hat{n}} \rangle}_{\beta_1} \\
&= \sum_{i=0}^{\ell} \sum_{j=0}^{\kappa} \hat{\alpha}_{i,j} B_i(z_1) C_j(z_2) + \beta_1,
\end{aligned}
\tag{5}
$$

where $\hat{\alpha}_{i,0} = w_i, \hat{\alpha}_{0,j} = w_{i+\ell}$ for all $i, j \geq 1$, for all other values of $i, j$ we set $\hat{\alpha}_{i,j} = 0$.

Moreover, by equation 2 we have that $\boldsymbol{U}_{1,:} = \sum_{i=1}^{\ell} \boldsymbol{v}_i B_i(z_1)$ and $\boldsymbol{U}_{2,:} = \sum_{i=1}^{\kappa} \boldsymbol{v}_{i+\ell} C_i(z_2)$, whereas the remaining rows $\boldsymbol{U}_{3,:}, \ldots, \boldsymbol{U}_{\hat{n},:}$ do not depend on $z_1, z_2$. First, let us rewrite the interaction between $z_1, z_2$

specifically:

$$
\begin{aligned}
\langle \boldsymbol{U}_{1,:}, \boldsymbol{U}_{2,:} \rangle_{M_{1,2}} &= \left\langle \sum_{i=1}^{\ell} \boldsymbol{v}_i B_i(z), \sum_{i=1}^{\kappa} \boldsymbol{v}_{i+\ell} C_i(z) \right\rangle_{M_{1,2}} \\
&= \sum_{i=1}^{\ell} \sum_{j=1}^{\kappa} \underbrace{\langle \boldsymbol{v}_i, \boldsymbol{v}_{j+\ell} \rangle_{M_{1,2}}}_{\gamma_{i,j}} B_i(z_1) C_j(z_2)
\end{aligned}
\tag{6}
$$

By following similar logic to 4, one can obtain a similar expression when looking at the partial sums that include the interaction between $f$ (resp. $e$) and all other fields except $e$ (resp. $f$). Observe that the interaction between all other values does not depend on $z_1, z_2$. Given all of this, we show how to rewrite the interaction as a function of $z_1, z_2$:

$$
\begin{aligned}
\sum_{i=1}^{\hat{n}} & \sum_{j=i+1}^{\hat{n}} \langle \boldsymbol{U}_{i,:}, \boldsymbol{U}_{j,:} \rangle_{M_{f(i),f(j)}} \\
&= \langle \boldsymbol{U}_{1,:}, \boldsymbol{U}_{2,:} \rangle_{M_{1,2}} + \sum_{j=3}^{\hat{n}} \langle \boldsymbol{U}_{1,:}, \boldsymbol{U}_{j,:} \rangle_{M_{1,f(j)}} \\
&\quad + \sum_{j=3}^{\hat{n}} \langle \boldsymbol{U}_{2,:}, \boldsymbol{U}_{j,:} \rangle_{M_{2,f(i)}} + \underbrace{\sum_{i=3}^{\hat{n}} \sum_{j=i+1}^{\hat{n}} \langle \boldsymbol{U}_{i,:}, \boldsymbol{U}_{j,:} \rangle_{M_{f(i),f(j)}}}_{\beta_2} \\
&= \sum_{i=1}^{\ell} \sum_{j=1}^{\kappa} \gamma_{i,j} B_i(z_1) C_j(z_2) + \sum_{i=1}^{\ell} \tilde{\alpha}_i B_i(z_1) \\
&\quad + \sum_{i=1}^{\kappa} \bar{\alpha}_i C_i(z_2) + \beta_2
\end{aligned}
$$

where $\tilde{\alpha}_i, \bar{\alpha}_i$ are obtained similarly in equation 4's final step. Consequently,

$$
\text{seg}[\Phi_{\text{FmFM}}^{\text{enc}}, 1, 2](\boldsymbol{z}_{-(1,2)})(z_1, z_2)
$$

is an affine function of the tensor-product basis $\{B_i(z_1) C_j(z_2)\}_{i,j=1}^{\ell,\kappa}$. $\qquad\square$

