# OpenReview forum: "Function Basis Encoding of Numerical Features in Factorization Machines"
_TMLR — Accepted by TMLR_

### Review · Reviewer_9Z8u · 2024-10-13

**Summary Of Contributions:**

This paper addresses feature preprocessing, particularly discretization of numerical features, to be used with factorization machines in the context of recommendation systems. The authors propose a basis function encoding of features, using B-splines basis functions, and evaluate their proposal on both tabular ML and recommendation system settings.

**Audience:**

Yes

**Claims And Evidence:**

No

**Requested Changes:**

A) The content needs to be reorganized and reordered. First describe what exactly you're doing, then later use proofs to argue about the properties of your proposal. If related work is going to just list related papers without explaining the research context, it should be moved towards the end. The experiments section should not meander into discussing how to address integer features; that belongs elsewhere in the paper.

B) Consider whether standard ML formalisms might be preferrable to RecSys terminology. For example, I suspect, but am not sure, whether "transformation -> segmentation -> field creation" is exactly equivalent to discretization with one-hot encoding.

C) As mentioned previously, there's a vast literature on feature normalization and discretization, and at least some standard baseline methods and current leading methods should be compared against. For normalization, standard baselines include sklearn's RobustScaler, and for discretization, standard baselines include uniform, quantile and k-means discretization. There is also finite scalar quantization [Mentzer, Fabian, et al. "Finite scalar quantization: Vq-vae made simple." arXiv preprint arXiv:2309.15505 (2023)], which in this setting is would involve applying rounding to normalized features to make them categorical. And there is kernel density integral (KDI) transformation [McCarter, C. The Kernel Density Integral Transformation. TMLR 2023.], which proposes adaptive methods for both normalizing features with skewed distributions and for discretizing the normalized features.

D) Figure 2 uses "x" notation to mean multiplication presumably, but is this a dot product, cross product, element-wise multiplication, or elementwise after broadcasting? This kind of mathematical sloppiness occurs elsewhere.

E) It is claimed but never explained (or empirically shown) why FMs benefit from discretization of numerical features.

F) Figure 4 -- why is there lag for orange vs blue? Is this a problem?

G) Section 4.2 -- please provide more clarity on the problem setup. For example, is this classification or regression?

**Strengths And Weaknesses:**

Strengths:

1. Much recent work on feature preprocessing focuses on tabular ML and/or time-series settings, so examining this problem in the RecSys setting is interesting.

2. As far as I'm aware, use of B-splines for feature preprocessing is original and interesting.

3. The authors provided code, which is very helpful.

Weaknesses:

1. The writing needs major revisions to improve clarity. The paper needs to more clearly delineate between previous work and the new contributions. The background should not assume total familiarity with factorization machines; the FM description in the appendix was necessary to even try to understand the main text. The contributions should state precisely what is being proposed. The implications of the proved spanning properties are not explained -- why do these properties matter?

2. If one is to evaluate a new method on tabular datasets, it is best practice to use a preexisting benchmark (eg OpenML or UCI++), instead of choosing one's own datasets (which can always be cherry-picked). Or, if one has particular reasons to choose particular datasets, the criteria for choosing these ones should be provided. (Granted, the authors seem to choose datasets with large N, which is probably justifiable as a proxy for RecSys scenarios; but this should be explained.)

3. Given the interaction between numerical feature normalization and the discretization that follows, it is not clear whether the empirical benefits of B-splines generalize to other normalizations than the ones used in the paper.

4. Other standard normalization and discretization methods should be compared against. (See below.)

---

> ### Author Response · Authors · 2024-10-23
> **Comments about requested changes**
>
> We thank the reviewer for reading the paper and their feedback. Below are our comments - some are disagreements where we explain why, and some are agreements. In both the disagreements and agreements we typically suggest solutions that might be an improvement upon the paper, since even if we disagree, there is a reason why the reviewer thought what they thought.
>
> A) The style of genericity --> specificity is a legimate style in scientific writing. i.e - first describe something about a generic basis and the corresponding properties, and then describe a concrete proposal of a spline basis and the corresponding properties.
>
> The general idea of "factorization machine of the FmFM family" + "basis" => "tensor product with factorized coefficients" stands in its own right. Even if the basis is *not* splines. Presenting the general idea first, and then studying the properties of the concrete choice of the B-Spline basis, to the best of our understanding, is a reasonable style of scientific writing.
>
> If you were referring to a different reorganization, please elaborate.
>
> B)
> **Segmentation**, to the best of our knowledge, is called [Currying](https://en.wikipedia.org/wiki/Currying) in mathematics, named after the logician Haskell Curry. Given a function $f(x, y)$, the function $f_x(y) = f(x, y)$, as a function of $y$, is a "Curried" version of $f$.In this case we believe it's better to use the RecSys term. i.e. $x$ is a given segment of users / items, and we care about understanding $f_x(y)$ - the behavior for that segment as a function of $y$. It's more insightful to us, and we believe to our audience as well. But we may be biased - we come from the RecSys community. If requested by the editor, we shall use the mathematical term of Currying.
>
> **Discretization**, to the best of our knowledge, is the mathematical term for taking a continuous curve and approximating it by a sequence of discrete points. Googling "curve discretization" yields plenty of literature. If you were referring to something different, please say so. See also this [Wikipedia article](https://en.wikipedia.org/wiki/Discretization). This is *not* RecSys terminology.
>
> **Fields** is the standard term used in literature on factorization machines. They refer to the columns of a table in tabular data - the columns are fields, whereas the values are features. i.e. Country is a field, whereas U.S or U.K are features. See papers cited from the first paragraph of Section 2.1. We do not believe using a different terminology than what's practiced in these papers is more insightful.
>
> **Transformation**, as you pointed out, is also called scaling or normalization. i.e. Standard**Scaler** / MinMax**Scaler** / Quantile**Transformer** / Power**Transformer** classes Scikit-Learn. We do believe that Scikit-Learn compliant terminology is well-known in the ML audience. We can use the term *scaling* if it's decided that it's more appropriate.
>
> C) Regarding feature scaling, here we agree with the reviewer. It may be a better approach to use the simple and known scaling / transformation techniques, such as MinMax or Quantile. We found that the CDF works great on our product, which essentially sparked the entire project of writing this paper, but for a scientific paper it may defocus it from the main idea. We wanted to share something that works well for us, but it might be the 'too much' that makes this paper less readable to a wider audience.
>
> D) Regarding Figure 2. We agree with the reviewer. The description says what it is: multiplying each row by the corresponding scalar, but it should also be apparent without reading the description. Maybe writing multiple "x" signs for each row, so that it's clear that each row separately is multiplied by a scalar, would be a better idea.  It can also be written as a matrix product of a diagonal matrix on the left, but we believe it would be more confusing this way.
>
> E) Indeed it is not explained. It is 'clear' to people working in FMs (it's a linear function of any given feature), but it needs to be explicitly said.
>
> F) We don't know. But looking at the code - we haven't found an issue.
>
> G) We treat all datasets as regression, i.e. predicting the conditional mean. This is true for both binary and non-binary labels. This coincides from the world we come from - CTR prediction. And this is where FMs are mostly used. But the reviewer might have pointed out an important aspect - we should either say it explicitly, or change the metric for binary cases to something like Accuracy, to that we also have classification tasks. Let the editor decide.

---

> > ### Comment · Reviewer_9Z8u · 2024-10-26
> > **Re: Comments about requested changes**
> >
> > Thanks for the thorough, helpful responses.
> >
> > (A) I agree with this overall organization approach for explaining the "what", but (first) I think this could be made more explicit, and (second) I think it's missing the "why". First, the "tensor product with factorized coefficients" is a major contribution towards scaling to many features, statistically and computationally, but this is not really highlighted in the paper (especially in the Introduction), and was not clear to me after my first reading. Note that "tensor product" only occurs twice (in Section 3.1 of the paper), and neither occurrence highlights the centrality of this to the proposed method.
> >
> > Second, as I asked in my review, "The implications of the proved spanning properties are not explained -- why do these properties matter?" As far as I understand it (please correct me if I'm wrong), the key problems you've solved arise from the fact that there are many features. The "many features problem" is why the segmentized orange line in Figure 1 (right) shoots far above and below the blue line; with a single field, presumably the orange line would be jagged, but would still stay close to the blue line. And then there is the computational problem of learning coefficients for interactions as the number of features grows. Basis functions address the computational problem (and this is what Section 3.1 is all about, IIUIC), while B-spline basis functions in particular address the statistical problem by providing smoothness. These are valuable solutions to key problems, but I had to infer them, because they were not explicitly spelled out to the reader. For example, one way to address this would be to motivate Lemmas 1 and 2 by summarizing their meaning and implications at the beginning of Section 3.1.
> >
> > (Moving the FM background from the Appendix to the main text, and explaining the benefits of discretizing numerical fields are my other main requests in this regard.)
> >
> > (B) Thanks for the clarifications. On reflection, I agree it does make sense to use RecSys terminology given that most readers will be more familiar with the RecSys terms. But I think including these clarifications in either footnotes or the appendix would be helpful for non-RecSys readers; in particular, "segmentation = currying" and "the columns are fields, whereas the values are features" were not obvious to me.
> >
> > (C) I think it's fine to focus on using the proposed approach with the CDF (eg in Section 4.4) if that's what empirically works best. But I suspect some potential users of your method will want to see results for a few other transformations on the datasets used in the Section 4.2, even if these results are best relegated to the appendix.
> >
> > (D-G) Thanks for the clarifications. Incorporating these small changes into the manuscript would address my concerns.
> >
> > Finally, a minor nitpick which I forgot to mention in my initial review: Some of the textual citations should instead be parenthetical citations, eg first paragraph of the Introduction.

---

> ### Author Response · Authors · 2024-10-23
> **Comments about weaknesses**
>
> 1. Indeed, we agree. The FM description should be pulled up from the appendix. It was supposed to be a conference paper, and this went there because of length limitations, but TMLR does not impose such strict limitations.
> 2. The datasets used in Section 4.1 were previously used for tabular benchmarks with NN models that attempted to improve the representation of numerical fields. We adopted the data-sets used by the works of Gorishny (2021, 2022), cited in the paper, which are NeurIPS papers on tabular data-sets with NNs that became relatively popular. We chose the subset of those data-sets that can work with a model that produces a scalar (binary labels / continuous labels), and discarded those that require producing a vector (i.e. multi-class labels, which require a logit vector). This is, of course, because a factorization machine produces a scalar value. *We agree with the reviewer* - saying directly in the paper *why* these data-sets were chosen would improve the credibility of the paper.

---

> > ### Comment · Reviewer_9Z8u · 2024-10-26
> > **Re: Comments about weaknesses**
> >
> > Thanks for the helpful clarifications; incorporating these into the manuscript will address my concerns.

---

### Review · Reviewer_oC7P · 2024-10-22

**Summary Of Contributions:**

This paper addresses the limitations of traditional Factorization Machine (FM) variants in recommendation systems that rely on binning numerical features. FM models are widely used in online content recommendation due to their balance between prediction accuracy, fast training, and inference times; however, the current binning-based approaches whose output is a step fundtion have difficulties encoding numerical features. These step functions often fail to capture the true, often smooth, nature of user behavior, leading to a gap between the model's learned functions and the actual behavior and increasing the number of bins is simply not effective. To deal with the existing issues, the paper propose a method that encodes the numerical features using a vector basis function while introducing a small change in the computation graph. They propose to use B-Splines which are particularly advantageous due to their strong approximation power and the fact that they require relatively few breakpoints (knots), minimizing the risk of sparsity. The expeirments are performed on both synthetic and real tabular datasets and demonstrate remarkable improvement.

**Audience:**

Yes

**Broader Impact Concerns:**

I don't have any broader impact concern.

**Claims And Evidence:**

Yes

**Requested Changes:**

I would expect to see more discussion about B-spline based methods in the literature.

**Strengths And Weaknesses:**

**Positive:**
- The paper motivates the problem and the proposed approach quite clearly.
- The datasets used in the experiments seem sufficient to show the advantage of the proposed method.


**Negative**
I am not an expert in the field. In general I liked the presented idea and how the paper is structured. However, I have some concerns.
- B-splines are extensively used for such function approximations in different fields, e.g., image processing and it is usually one of the first choices. I am a bit surprised that a similar idea was not presented before. Since I am not familiar with the literature, I cannot comment about the novelty but I would strongly advise authors to check the relevant literature and compare with spline based methods. For example, the following reference I found with a quick search seems relevant [1].

[1] Lan et al, Accurate and Interpretable Factorization Machines, AAAI 2020.

---

> ### Author Response · Authors · 2024-10-23
> **Prior art**
>
> Spline based methods are indeed quite widely used, but the focus of this paper is on the interplay between splines and factorization methods. We found only one line of work, that of Rugamer, and cited it in the related works section. Splines, of course, are extremely widely used in ML and outside of ML in a huge variety of fields. If you believe that citing more works on splines would benefit, we will, but we believe it's out of the focus of this paper. In this case, a good comprimise would be to cite very prominent ML papers with splines (such as MARS regression), and a book where the subject is elaborated.
>
> The paper you referred to, by the way, does not appear to use splines anywhere.

---

### Review · Reviewer_1MX4 · 2024-10-24

**Summary Of Contributions:**

This paper proposes a new technique for incorporating numerical features into Factorization Machine (FM) variants, which are widely used in large-scale real-time content recommendation systems. The proposed method encodes numerical features into a vector of function values for a set of functions of one's choice, allowing the model to learn segmentized functions of the numerical feature spanned by the chosen basis. The authors demonstrate the effectiveness of their approach through experiments on synthetic and real-world datasets

**Audience:**

Yes

**Claims And Evidence:**

No

**Requested Changes:**

Illustrate the issue of using the numerical columns naively in the FM.

**Strengths And Weaknesses:**

[Strengths]
* FM is an important technique, and the problem addressed in the paper is relevant to the field. The authors tackle the problem of effectively incorporating numerical features into FM models.
* The presentation is clear, with a comprehensive evaluation of the proposed approach. Empirical results suggest benefits of the proposed method. The code and results are also made publicly available, allowing for easy reproduction and verification.
* The encoding scheme using basis functions has potential benefits if the basis functions are carefully chosen for the specific data. The authors demonstrate that their approach can improve the accuracy of FM models.

[Weaknesses]
* Motivation: it is not entirely clear why numerical columns could pose an issue for the FM. The authors could provide more context or examples to illustrate the limitations of using numerical columns naively in FM, and how their proposed approach addresses these limitations.
* Novelty of approach: The proposed approach seems to be a feature pre-processing/transformation step before the FM, rather than a fundamental change to the FM formulation itself. While the use of basis functions may have some benefits, it is not clear whether this approach is significantly different from other feature transformation techniques that have been proposed in the literature.

---

> ### Author Response · Authors · 2024-10-27
> **Comment**
>
> We thank the reviewer for reviewing the paper, and have the following comments about the weaknesses:
> - Motivation: without any feature preprocessing, an FM is just a linear function of any _one_ of the features, assuming the remaining are kept constant. So even _binning_, which is the standard approach used with FMs in practice in most cases, and also won the Criteo and Avazu Kaggle challenges, exists solely to represent a non-linear function. We agree with the reviewer that this should be explicitly said in the paper, even if it may be 'obvious'
> - Novelty: eventually, we're typically interested in what a model _including_ any feature preprocessing, represents as a function of its _original_ inputs. For example, polynomial regression is just feature preprocessing on  top of linear regression, but it represents a polynomial in terms of the original input. The interplay between basis encoding and the "field-informed" factorization machine variants has not been studied previously, to the best of our knowledge. The interplay between basis encoding and regular FMs has been, and we cite it in related work. Our work formalizes this interplay, and explains why it's useful in recommender systems. We believe it is both interesting theoretically, because an extremely simple argument produces an interesting and useful outcome, and also valuable in practice to some researchers working on FMs, especially in the industry. This is true especially in the industry, methods that require only a small incremental change to an existing system, and are simple to implement, are valuable.

---

### Author Response · Authors · 2024-11-30
**Revision**

I added a revision to the paper to address the main reviewer comments. Here is a short summary:
- pulled FM background from the appendix, and created a 'background' section.
- formally defined what 'segmentized' means in the background section using a $\mathrm{seg}$ opeator, and use it throughout the paper.
- explained why binning is typically used with FM models in the background section.
- clarified that fitting a parametric distribution is just an efficiency enhancement over the well-known quantile transform
- in the experimental section, clarified why the data-sets were chosen, and that the task is regression.
- wrote in the intro section that we're getting factorized tensor-product basis, and also repeated a more elaborate explanation in the section about spanning properties.
- clarified in the intro section that our aim is analysis of the interplay between basis functions and factorization machines, and not proposing a new way to encode numerical fields with basis functions.
- re-draw Figure 2 to show that x multiplies all rows of the matrix V that contains the embedding vectors.

On the one hand, nothing of substance changed - the claims are the same claims, the results are the same results, and the experiments are the same experiments. However, the text became longer and more ``formal''. But I hope it improves the quality of the paper to your satisfaction.

---

> ### Comment · Reviewer_9Z8u · 2024-12-05
> **Re Revision**
>
> I appreciate the work into these updates. My main concerns were addressed to my satisfaction.

---

### Decision · Action_Editor_1ymB · 2024-12-16

**Recommendation:** Accept with minor revision

**Comment:**

The revisions made by the authors have improved the paper substantially and addressed the reviewer concerns.

Please make the following changes before submitting the final reivison:
- Proofread the paper, there is still a number of typos in it.
- Make sure that citations use parenthesis when appropriate.
- Define 1_l notation in Lemma 1.
- Remove "why?" from Section 5.

**Audience:**

Using splines to increase the modelling power of Factorization Machines on numerical fields is of interest to some readers of TMLR.

**Claims And Evidence:**

Yes

---

> ### Author Response · Authors · 2024-12-22
> **Camera ready uploaded**
>
> The camera ready revision that includes the requests by the AE has been uploaded.
>
> We would like to thank the reviewers and the AE for their valuable feedback that made this paper better and more rigorous.